# Role of Clinical Characteristics and Biomarkers at Admission to Predict One-Year Mortality in Elderly Patients with Pneumonia

**DOI:** 10.3390/jcm11010105

**Published:** 2021-12-25

**Authors:** Astrid Malézieux-Picard, Leire Azurmendi, Sabrina Pagano, Nicolas Vuilleumier, Jean-Charles Sanchez, Dina Zekry, Jean-Luc Reny, Jérôme Stirnemann, Nicolas Garin, Virginie Prendki

**Affiliations:** 1Department of Rehabilitation and Geriatrics, Division of Internal Medicine for the Aged, University Hospitals of Geneva, Hôpital des Trois-Chêne, 1226 Thonex, Switzerland; dina.zekry@hcuge.ch (D.Z.); virginie.prendki@hcuge.ch (V.P.); 2Department of Internal Medecine, Medical Faculty, Geneva University Hospitals, 1205 Geneva, Switzerland; leire.azurmendi@unige.ch (L.A.); sabrina.pagano@hcuge.ch (S.P.); Nicolas.vuilleumier@hcuge.ch (N.V.); jean-charles.sanchez@unige.ch (J.-C.S.); 3Diagnostic Department, Division of Laboratory Medicine, Geneva University Hospitals, 1205 Geneva, Switzerland; 4Medical Faculty, University of Geneva, 1211 Geneva, Switzerland; jean-luc.reny@hcuge.ch (J.-L.R.); Nicolas.garin@hcuge.ch (N.G.); 5Department of Internal Medicine, Division of General Internal Medicine, Geneva University Hospitals, 1205 Geneva, Switzerland; jerome.stirnemann@hcuge.ch; 6Department of General Internal Medicine, Riviera Chablais Hospitals, 1847 Rennaz, Switzerland; 7Department of Internal Medicine, Division of Infectious Diseases, Geneva University Hospitals, 1205 Geneva, Switzerland

**Keywords:** elderly patients, one-year mortality, biomarkers

## Abstract

Background: A hospitalization for community-acquired pneumonia results in a decrease in long-term survival in elderly patients. We assessed biomarkers at admission to predict one-year mortality in a cohort of elderly patients with pneumonia. Methods: A prospective observational study included patients >65 years hospitalized with pneumonia. Assessment of PSI, CURB-65, and biomarkers (C-reactive protein (CRP), procalcitonin (PCT), NT-pro-B-type natriuretic peptide (NT-proBNP), interleukin (IL)-6 and -8, tumor necrosis factor alpha (TNF-α), serum amyloid A (SAA), neopterin (NP), myeloperoxidase (MPO), anti-apolipoprotein A-1 IgG (anti-apoA-1), and anti-phosphorylcholine IgM (anti-PC IgM)) was used to calculate prognostic values for one-year mortality using ROC curve analyses. Post hoc optimal cutoffs with corresponding sensitivity (SE) and specificity (SP) were determined using the Youden index. Results: A total of 133 patients were included (median age 83 years [IQR: 78–89]). Age, dementia, BMI, NT-proBNP (AUROC 0.65 (95% CI: 0.55–0.77)), and IL-8 (AUROC 0.66 (95% CI: 0.56–0.75)) were significantly associated with mortality, with NT-proBNP (HR 1.01 (95% CI 1.00–1.02) and BMI (HR 0.92 (95% CI 0.85–1.000) being independent of age, gender, comorbidities, and PSI with Cox regression. At the cutoff value of 2200 ng/L, NT-proBNP had 67% sensitivity and 70% specificity. PSI and CURB-65 were not associated with mortality. Conclusions: NT-proBNP levels upon admission and BMI displayed the highest prognostic accuracy for one-year mortality and may help clinicians to identify patients with poor long-term prognosis.

## 1. Introduction

With an incidence between 1.2 and 11.6 cases for 1000 person per year, community-acquired pneumonia (CAP) is a serious public health problem with a significant morbidity, mortality, and economic burden in Europe [1]. CAP is often a turning point in the life of elderly patients, leading to loss of autonomy, cognitive decline, decompensation of comorbidities, and death. Rates of hospitalization are high, with prolonged hospital stays and a high mortality [2]. A hospitalization for CAP results in a decrease in long-term survival in elderly patients [3], partly attributed to the high burden of inflammatory stress, contributing to a high incidence of cardiovascular events after pneumonia [4,5]. Hence, biomarkers of inflammation and of cardiac impairment may help refine the prediction of long-term CAP prognosis in elderly patients [6,7].

Among traditional biomarkers of inflammation, C-reactive protein (CRP) and procalcitonin (PCT) are the most frequently used for prognostication of pneumonia in adults [8]. Serum amyloid A protein (SAA), neopterin (NP), myeloperoxidase (MPO), anti-apolipoprotein A-1 IgG (anti-apoA-1), and anti-phosphorylcholine IgM (anti-PC IgM) have been recently investigated in the same context and/or in cardiovascular diseases [9,10,11,12,13]. An association between high levels of pro and anti-inflammatory cytokines (especially interleukin (IL)-1, -6, and -11) with short- and long-term mortality of CAP has been described, but information concerning elderly patients is lacking [14,15,16,17,18].

The two most used prognosis scores for predicting pneumonia one-month mortality are the Pneumonia Severity Index (PSI) and Confusion, Urea, Respiratory Rate, Blood Pressure, and Age >65 years (CURB-65) criteria [19,20]. Both combine one or more biomarkers with demographic data and physiological measures, but they have limitations in elderly patients [21,22]. Hence, we aimed to test whether cardiovascular and inflammation biomarkers measured at hospital admission were associated with one-year mortality in elderly patients.

## 2. Materials and Methods

### 2.1. Design, Setting, and Population Study

This prognostic study used data collected in a prospective observational study [23]. Consecutive patients hospitalized with suspected pneumonia in the geriatric and internal medicine wards of Geneva University Hospitals, an 1800-bed tertiary-care institution serving a population of 500,000 inhabitants, were included between 1 May 2015 and 30 April 2016. Eligible patients were older than 65 years and presented symptoms and signs suggestive of a respiratory infection warranting antibiotic treatment. A confirmed diagnosis of pneumonia required demonstration of an acute lung infiltrate on low-dose CT scan (LDCT) and was adjudicated a posteriori by a panel of eleven senior physicians (experts) with expertise in the diagnosis and management of pneumonia and radiologists experienced in thoracic imaging. Experts analyzed clinical, biological, microbiological, and radiological data and patients’ outcomes and rated the probability of pneumonia according to a Likert scale (low, intermediate, or high). The reference diagnosis was considered positive (respectively, “negative”) if the panel of experts rated the probability of pneumonia “intermediate” or “high” (respectively, “low”) on the Likert scale. Patients who had been treated for pneumonia during the previous 6 months or admitted to the intensive care unit were not included. Patients’ management, including the choice and duration of antibiotic therapy, followed local guidelines. The study was approved by Geneva’s Institutional Review Board (CER-14-250), and it was registered at clinicaltrials.gov (NCT02467192). Written informed consent was obtained from all patients or next of kin.

### 2.2. Data Collection

Demographic data, comorbidities, anamnestic data, vital signs, physical signs, severity scores (PSI, CURB-65), and the results of routinely obtained blood tests (including CRP, PCT, and NT-pro-brain natriuretic peptide (NT-proBNP)) were recorded at admission [19,20].

### 2.3. Biomarkers Measurement

Plasma CRP concentrations were measured via immunoturbidimetry (Roche/Hitachi Cobas c702 systems) and PCT using a rapid assay with a sensitivity of 0.06 µg/L (Kryptor PCT, Brahms, Hennigsdorf). Blood was sampled for IL-6, IL-8, TNF-α, SAA, NP, MPO, anti-apoA-1 IgG, and anti-PC IgM levels determination within 48 h after admission. Measurements were performed after completion of the study (Appendix B).

### 2.4. Outcome

Survival status and date of death up to one year were assessed by searching a posteriori in medical records and the register of deaths by an investigator blinded to the measurements of the investigational biomarkers.

### 2.5. Data Analysis

Sample size was determined based on the power calculation of the original study [23]. We used frequencies, percentages, mean with range, and median with interquartile range for descriptive purposes. Variables were compared between patients dead or alive at one year in univariate analysis using One-way Analysis of Variance (ANOVA) for continuous variables and Fisher’s exact test or chi-square test for categorical variables, as appropriate. The discrimination ability of each biomarker for one-year mortality was assessed by computing the area under the receiver operating characteristic curve (AUROC) with 95% CI. Univariate Cox regression was used to obtain the hazard ratio (HR) of death by individual predictive variables. Variables associated with the outcome on univariate analysis with a *p* value < 0.10 were entered in a multivariate Cox proportional hazard model using backward conditional regression. PSI and gender were forced in the model, as they are well-described prognostic factors. No imputation was made for missing values.

Optimal cutoff for biomarkers significantly associated with the outcome in multivariate analysis was determined with the Youden index. We plotted Kaplan–Meyer survival curves at optimal cutoff and computed sensitivity, specificity, and positive and negative predictive values. The study sample was determined by the design of the original study. All *p* values are based on two-tailed tests and are considered significant for *p* < 0.05. Data were analyzed using SPSS version 25 (IBM SPSS Statistics for Windows, Version 25.0. Armonk, NY: IBM Corp. Released 2017).

## 3. Results

### 3.1. Clinical Characteristics

Of 200 patients included in the original study, 133 patients (median age 83 years (IQR: 78–89)) had confirmed pneumonia, of whom 35 (26%) died within one year. Table 1 displays the main characteristics of the patients according to one-year mortality. Sex ratio was approximately 1. One hundred and seventeen patients (88%) lived at home. Main comorbidities were heart failure (*n* = 30, 23%), chronic pulmonary disease (*n* = 29, 22%), dementia (*n* = 27, 20%), and chronic renal failure (*n* = 24, 18%). Most patients presented with cough (*n* = 120, 90%), dyspnea (*n* = 95, 71%), and lung crackles (*n* = 114, 86%).

### 3.2. Biomarkers Results and Outcome

Table 2 shows the levels of biomarkers at admission according to the outcome and the areas under the receiver operating characteristic curves (AUROCs) for mortality prediction. Data were complete for CRP. There were 18 missing data for NT-proBNP (14%), 12 for PCT (9%), and 3 (2%) for each other biomarker. Only NT-proBNP and IL-8 showed significant discrimination.

The univariate and multivariate associations of each biomarker with one-year mortality are shown in Table 3. In univariate analysis, older age, lower BMI, and dementia were associated with a higher risk of death. Among biomarkers, NT-proBNP and IL-8 were associated with death. In a multivariate model adjusting for age, BMI, dementia, IL-8, and NT-proBNP, only NT-proBNP (HR = 1.01 per 100 ng/mL (CI 95% 1.00–1.02), *p* = 0.05) and BMI (HR 0.92 (CI 95% 0.85–1.00), *p* = 0.05) were associated with one-year mortality.

The optimal cutoff was 2200 ng/L for NT-proBNP at admission, with a sensitivity of 67% and a specificity of 70% for one-year mortality (for a normal range of <300 ng/L). Figure 1 displays survival curves for NT-proBNP.

Neither CURB-65 (AUROC = 0.57 (CI 95% 0.45–0.69), *p* = 0.23) nor PSI score (AUROC = 0.59 (CI 95% 0.48–0.70), *p* = 0.12) were associated with one-year mortality.

In a post hoc analysis, we tested whether the association of BMI and NT-proBNP with mortality remained significant after adjusting for the markers of renal function creatinine and urea. When urea was added in the model, aHR of BMI and NT-proBNP did not change; the same was observed after adding creatinine instead of urea (results not shown).

## 4. Discussion

In this cohort of elderly patients hospitalized for pneumonia, only BMI and NT-proBNP were independently associated with one-year mortality. Neither usual pneumonia severity scores (PSI and CURB-65) nor any biomarker of inflammation were predictive of one-year mortality.

One-year mortality was 26%, which is coherent with the known poor prognosis of elderly patients after pneumonia [7,16,24]. Bruns et al. showed that long-term mortality in patients recovering from CAP was more than 3 times higher than in the general population, and the most frequent causes of death were related to co-morbidities, including malignancy (27%), chronic obstructive pulmonary disease (19%), and cardiovascular disease (16%) [25].

Higher NT-proBNP is a powerful predictor of death in a range of conditions. NT-proBNP is released from cardiac myocytes in response to mechanical load and wall stress. NT-proBNP is not solely a marker of impaired left ventricular systolic function but also of abnormalities of diastolic dysfunction, right ventricular dysfunction, valvular dysfunction, increased pulmonary pressures, and atrial arrhythmias. It has been identified not only as a prognostic biomarker in pulmonary arterial hypertension, myocardial infarction, valvular heart disease, atrial fibrillation, and pulmonary embolism but also as an identifier of sepsis patients at high risk for functional decline [26,27,28]. Hence it is not surprising that NT-proBNP was an independent predictor of mortality in this cohort of elderly patients hospitalized for pneumonia. Heart failure and other cardiovascular diseases are frequent conditions in patients hospitalized for CAP, and CAP itself can accelerate the pathological process through destabilization of the vascular endothelium and acceleration of the progression of atherosclerosis [5,29]. Eurich et al. showed that CAP increased the risk of heart failure, and some authors have stated that pneumonia itself should be considered an independent cardiovascular risk factor leading to acute myocardial infarction, cardiogenic edema, arrhythmia, and stroke [4,30,31]. Menendez et al. reported that cardiac biomarkers, especially NT-proBNP, can identify patients with CAP at high risk for early and long-term cardiovascular events [24]. A score named the UBMo index developed in a cohort of elderly patients (mean age of 85 years) and incorporating NT-proBNP, urea, and monocyte count was very efficient in identifying patients at high risk of one-year mortality after a pneumonia [7]. An elevated NT-proBNP at admission in an elderly patient hospitalized for pneumonia should prompt consideration of active management of known or latent heart disease, including strengthened follow-up after discharge.

Pneumonia severity scores—namely, PSI and CURB-65—have been developed to predict the risk of mortality at 30 days, in order to guide therapeutic management [19]. Both scores are predictive of the mortality risk up to 6 years after admission in younger patients [32]. However, they may be less accurate in elderly patients. Putot et al. showed that PSI but not CURB-65 correlated with one-year mortality, and their new index named UBMo performed better than PSI score in predicting one-year mortality [7]. In our cohort, no score correlated with mortality. CURB-65 is known to be inaccurate in older patients, as it does not take into account comorbidities and may underestimate the risk of death [33].

We showed that lower BMI was associated with an increased risk of death. Malnutrition is multifactorial, closely related with frailty and comorbidities, and is already known to be associated with a poor long-term outcome in patients with CAP, particularly the elderly [34,35]. As it is amenable to nutritional interventions, malnutrition should be thoroughly evaluated in elderly patients hospitalized for CAP.

Measuring inflammation is theoretically an attractive target for long-term prognostication. Chalmers et al. showed that high levels of CRP at admission and at 3–4 days are related to an increased risk of complications and short-term mortality in CAP [36]. However in a recent publication, increased CRP levels failed to predict the occurrence of cardiovascular events in the short or long term in CAP patients [24]. In our cohort of elderly patients, we did not find that CRP level at admission was associated with one-year mortality. In a similar population, Karasahin et al. described that monitoring changes in serial CRP measurements could be useful for prognosis in elderly patients with infection [37]. They showed that a CRP decrease less than 11% over 48 h and less than 20% over 96 h was a risk factor for mortality. The literature frequently assessed the association of initial inflammation and in-hospital mortality, as for example, the association of higher PCT and mortality in pneumonia [8]. However, initial values may be insufficient for long-term prognostication, and assessment of the evolution of inflammation should be more relevant.

As for newer biomarkers, SAA is correlated with CRP in CAP [38]. Measurement of SAA in patients with ventilation-acquired pneumonia had a good predictive value for mortality [9]. NP is a marker of cellular immunity associated with the severity of infection and mortality in critically ill patients [10,39]. In chronic obstructive pulmonary disease (COPD), high NP levels during an exacerbation episode correlated with short-term prognosis [8]. Anti-apo-A1 is a mediator of inflammation and atherosclerosis, correlated with inflammation, endothelial dysfunction, and rupture of atheromatous plaques [40]. Anti-PC IgM is part of the innate immune system and is known as a major factor in the defense against pneumococcus [41,42]. Anti-PC IgM also prevents the development of atheromatous plaque, inhibiting the exit of OxLDL and the PAF [43]. MPO is an enzyme released by activated neutrophils during an infection [13]. In the ICU setting, MPO levels were related to 30-day mortality, and high MPO levels increased mortality risk on top of the APACHE IV score [12]. Unfortunately, we could not show any link between these newer biomarkers and long-term survival in our cohort.

Pinargote-Celorio et al. recently confirmed previous findings suggesting that pro-inflammatory cytokines (IL-6, IL-8) and anti-inflammatory cytokines (IL-10) could predict one-month mortality for CAP in elderly patients [14,18]. IL-6 is a cytokine of the acute phase and has been mostly described as a risk factor for early mortality [15,44]. IL-8 is an inflammatory chemokine that plays a key role in the recruitment and activation of neutrophils during inflammation, in atherogenesis, atherosclerotic plaque destabilization, neovascularization, and angiogenesis [45,46]. It has been shown that higher concentrations of IL-8 at admission were associated with in hospital mortality in ICU patients with severe CAP or sepsis [47,48]. Of note, few studies have been performed in elderly patients. In our study, IL-6 and IL-8 on admission were not associated with long-term mortality.

Our study is an original study and has several strengths. It included consecutive elderly patients with multiple comorbidities who had extensive diagnostic testing. A robust reference standard was used, based on CT scan findings. According to recent findings, a gold standard based on chest X-ray may lead to frequent misclassifications [23]. Moreover, we tested many inflammatory and cardiac biomarkers at admission.

Conversely, our study has some limitations. It is a single-center study with a small sample size. Some patients received previous antibiotics that may have altered biomarker levels and performance. Another limitation is that biomarkers are not static, and their measurement at a specific time does not give a complete picture of the condition. Finally, prior host status, severity of the disease, or other factors can also influence biomarkers.

## 5. Conclusions

In conclusion, a lower BMI and a higher NT-proBNP level at admission were independently associated with one-year mortality after a hospitalization for pneumonia in elderly patients. Destabilization of cardiovascular known or latent disease may be an important cause of mortality in this population, and a more accurate prognostication may be achieved through cardiac biomarker measurement, potentially allowing personalization of subsequent management. A thorough nutritional assessment should be performed, and nutritional interventions should be considered when relevant. These results need to be evaluated in larger and multicenter trials.

## Figures and Tables

**Figure 1 jcm-11-00105-f001:**
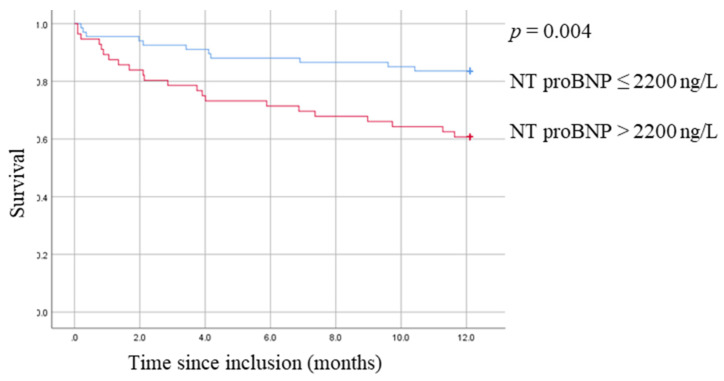
Survival curve of NT pro-BNP value at admission.

**Table 1 jcm-11-00105-t001:** Baseline characteristics of the 133 patients according to one-year mortality.

Characteristics	No (%) or Mean (SD)			*p* Value
	Overall (133)	Dead 35 (26)	Alive 98 (74)	
Demographics				
Female gender	60 (45.1)	15 (42.9)	45 (45.9)	0.76
Age (years)	82.9 (7.9)	85.5 (7.4)	82.0 (7.9)	**0.02**
Smoker (active or past)	80 (60.2)	19 (54.3)	61 (62.2)	0.41
BMI (kg/m^2^)	24.4 (5.1)	22.7 (5.4)	24.5 (4.9)	**0.03**
Place of living				0.14
Home	117 (88.0)	30 (85.7)	87 (88.8)	
Nursing home	11 (8.3)	5 (14.3)	6 (6.1)	
Other	5 (3.8)	0	5 (5.1)	
Hospitalized during last 6 months	39 (29.3)	12 (27.6)	27 (34.3)	0.45
Comorbidities				
Charlson score		3.25 (2.4)	3.0 (1.7)	0.52
Past myocardial infarction	22 (16.5)	8 (22.9)	14 (14.3)	0.24
Heart failure	30 (22.6)	8 (22.9)	22 (22.4)	0.96
Dementia	27 (20.3)	11 (31.4)	16 (16.3)	0.06
Chronic pulmonary disease	29 (21.8)	8 (22.9)	21 (21.4)	0.86
Diabetes	7 (5.3)	3 (8.6)	4 (4.1)	0.31
Chronic renal disease	24 (18.0)	5 (14.3)	19 (19.4)	0.50
Active cancer	7 (5.3)	3 (8.6)	4 (4.1)	0.31
Prognostic scores				
CURB-65	2.2 (0.9)	2.4 (1.0)	2.2 (0.8)	0.15
PSI	107 (26)	113 (28)	105 (25)	0.63
Vital signs				
Heart rate	93 (18)	95 (19)	92 (18)	0.41
Respiratory rate	25 (7)	24 (6)	25 (7)	0.53
Temperature (°C)	38.0 (1.0)	37.8 (1.0)	38.1 (1.0)	0.10
Systolic blood pressure (mmHg)	131 (23)	133 (26)	130 (22)	0.53

Abbreviations: BMI, body mass index; CURB-65, confusion, urea >7 mmol/L, respiratory rate ≥30 breaths/min, blood pressure <90 mmHg systolic or ≤60 mmHg diastolic, age ≥65; PSI, Pneumonia Severity Index.

**Table 2 jcm-11-00105-t002:** Biomarkers levels at admission according to one-year mortality, and area under the receiver operating characteristic curve.

	At Admission			*p* Value *	AUROC	*p*-Value **
	Overall (133)	Dead (35)	Alive (98)			
NT-proBNP (ng/L)	1826 (666–3917)	2998 (1459–5971)	1697 (595–3083)	<0.01	0.65 (0.54–0.77)	0.02
CRP (mg/L)	107 (58–208)	82 (43–159)	111 (58–223)	0.10	0.44 (0.33–0.54)	0.27
PCT (μg/L)	0.36 (0.14–1.99)	0.57 (0.12–2.49)	0.36 (0.14–1.93)	0.85	0.52 (0.40–0.64)	0.71
IL-6 (pg/L)	8.7 (3.7–19.4)	8.9 (3.9–18.3)	8.6 (3.5–19.6)	0.98	0.49 (0.38–0.60)	0.86
IL-8 (pg/L)	17.3 (11.1–26.5)	22.8 (14.1–29.8)	15.0 (9.5–23.3)	0.04	0.66 (0.56–0.75)	<0.01
MPO (pg/mL)	204.2 (116.9–408.2)	171.2 (119.5–297.1)	206.8 (114.5–473.7)	0.10	0.44 (0.34–0.55)	0.33
TNF-α (pg/mL)	3.2 (2.5–4.8)	3.2 (2.3–4.9)	3.3 (2.5–4.7)	0.17	0.48 (0.36–0.60)	0.70
SAA (μg/L)	268 (252–288)	276 (243–293)	267 (254–288)	0.44	0.50 (0.38–0.62)	0.98
anti-apoA-1 IgG (DO)	0.30 (0.18–0.47)	0.30 (0.20–0.50)	0.30 (0.20–0.40)	0.89	0.51 (0.39–0.63)	0.87
anti-PC-IgM (U/mL)	36.3 (20.3–61.5)	38.1 (21.4–62.9)	36.3 (20.0–62.3)	0.53	0.52 (0.40–0.63)	0.75
NP	6.6 (4.5–13.0)	5.9 (4.4–17.2)	7.1 (4.5–12.6)	0.86	0.48 (0.37–0.60)	0.78

Medians with interquartile range; AUROC with 95% CI; * Death vs. alive, ** Null hypothesis, true area = 0.5; Abbreviations: CRP, C-reactive protein; PCT, procalcitonin; IL, interleukin; MPO, myeloperoxidase; SAA, serum amyloid A; Anti-apo-A1, anti-apolipoprotein A-1; anti-PC IgM, anti-phosphorylcholine IgM; NP, neopterin; DO, optical density; AUROC, operating curve for mortality prediction.

**Table 3 jcm-11-00105-t003:** Univariate and multivariate association of biomarkers and clinical characteristics with one-year mortality at admission.

	Univariate Analysis		Multivariate Analysis	
	HR (CI 95%)	*p*-Value	HR (CI 95%)	*p*-Value
Age (/year)	1.05 (1.01–1.10)	0.02		
Gender (female)	0.94 (0.48–1.84)	0.87		
PSI score	1.01 (1.00–1.02)	0.09		
BMI (/kg/m^2^)	0.92 (0.85–0.99)	0.02	0.92 (0.85–1.00)	0.05
Dementia	2.14 (1.05–4.38)	0.04		
NT-proBNP (/100 ngl/L)	1.01 (1.00–1.02)	0.04	1.01 (1.00–1.02)	0.05
IL-8 (/pg/L)	1.01 (1.00–1.01)	0.01		

Abbreviations: BMI, body mass index; IL, interleukin.

## Data Availability

The data presented in this study are available on request from the corresponding author.

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
