# Peer review of "Role of Clinical Characteristics and Biomarkers at Admission to Predict One-Year Mortality in Elderly Patients with Pneumonia"

_jcm, 2021, doi:10.3390/jcm11010105_

Round 1

Reviewer 1 Report

This study, which is an ancillary study from a prospective observational study, aims to correlate long-term prognosis of CAP occurring in >65y old patients (n=133 patients of whom 35 died within 1 year) with several clinical and biological markers. Whereas in univariate analysis, age, BMI, dementia, NT-proBNP and IL-8 are associated with prognosis, only BMI and NT-proBNP levels are associated with one-year survival in the multivariate analysis. The manuscript is well written and presented.

My main concerns are:

1/ NT-proBNP levels might be elevated in case of renal failure.  Did the authors considered it and could they confirmed that NT-proBNP might be a prognostic marker independent of patient’s renal function?

2/ Although statistically significant, the NT-proBNP cut-off value of 2200 ng/mL seems not very discriminant for current practice.

  • Considering the important cardiovascular risk following CAP, especially in elderly patients, shouldn’t the search for potential underlying cardiac disease be generalized for every elderly patients recovering from CAP whatever NT-proBNP values?
  • Did previous study evaluated the potential long-term prognostic impact of a systematic screen for cardiovascular disease in such (or younger) population?
  • Did the authors searched for a composite score (including age, BMI, IL-8, dementia?) that could better discriminate patients with a poorer prognosis? Including an evaluation of coronary/vascular calcifications on Chest CT could also be interesting as part of this prognostication.

3/ If available, It would be interesting to precise the cause of deaths observed in this cohort (cardiovascular events?).

4/ In the abstract:

  • The “Background” section only contains a sentence that refers on the aim of the study.
  • In the “Methods” section, please correct “obseravtional “.

Reviewer 2 Report

In this study, authors found that high NT-proBNP level was an independent prognostic factor in elderly patients admitted with pneumonia.  This is an interesting study with good clinical significance.  The paper was well written with straightforward manner.  I have only minor comments.

Comments

  1. Although NT-proBNP level at admission was revealed to be a prognostic factor, there was no difference in heart failure and chronic renal disease complication between dead and alive patients as shown in Table1.  Please explain the reason for that in the discussion section.
  2. If it is possible, I would like to know the cause of death in 35 dead patients.

Reviewer 3 Report

Dear Authors

I've appreciated your paper; your data and your research are interesting. 

 Nevertheless, I think that you should better explain why clinical scores as CURB 65 and PSI don’t correlate with mortality in an elderly population affected by pneumonia, whereas a NT proBNP's single value does. 

In my opinion this could be due to the small sample size and “one center” study feature; in effect these are the main limits of your research. 

So in “discussion” or “ Conclusion” item you could insert a sort of starting point for future multicenter trials to have a high patients’ number, and available biomarkers values’ trends. 

In following items you could find some other requests : 

  • Line 38 Tab 1: Unexpectedly being a smoker (active or in past) wasn’t a risk-factor for outcome. How do you comment on this data: maybe in elderly patients this behavior is less linked to cardiovascular or pulmonary diseases? 
  • Line 38 Tab 1: I think that you should better explain the “Place of living” role. Usually living in a nursing home  could be a risk factor for mortality for several reasons, such as  higher comorbidities and the pneumonia’s nosocomial feature. In your research the rate of these patients is very low (8%) compared to living home ones (88%). So I think that this unusual distribution could be the reason for your not statistically significant p value of 0.14.
  • Line 149 Tab 2: in my opinion you should insert the NT-proBNP “normal range” value to have a better reference term. 
  • Line 218 I wonder if the low power of a clinical score as CURB 65 could be explained by the high rate of “living home” patients in your research.  

Best regard

Round 2

Reviewer 3 Report

Dear Authors,

I've read your revised manuscript. I've appreciated all the corrections and the improvements added in this new version.

I also have examined your responses to my requests in previous reviewer's reports.

I am satisfied; and in my opinion your manuscript could be edited, even considering the "small sample size" and "one-center study" limit.

Best regards